# Revealing Dominant Eigendirections via Spectral Non-Robustness Analysis in the Deep Reinforcement Learning Policy Manifold

## Abstract

Deep neural policies have recently been installed in a diverse set of settings, from biotechnology to automated financial systems. However, the utilization of deep neural networks to approximate the state-action value function commences concerns on the decision boundary stability, in particular, with regard to the sensitivity of policy decision making to indiscernible, non-robust features due to highly non-convex and complex deep neural manifolds. These concerns constitute an obstruction to understanding the reasoning made by deep neural policies, and their foundational limitations. Thus, it is crucial to develop techniques that aim to understand the sensitivities in the learnt representations of neural network policies. To achieve this we introduce a method that identifies the dominant eigen-directions via spectral analysis of non-robust directions in the deep neural policy decision boundary across both time and space. Through experiments in the Arcade Learning Environment (ALE), we demonstrate the effectiveness of our spectral analysis algorithm for identifying correlated non-robust directions, and for measuring how sample shifts remold the set of sensitive directions in the neural policy landscape. Most importantly, we show that state-of-the-art adversarial training techniques yield learning of sparser high-sensitivity directions, with dramatically larger oscillations over time, when compared to standard training. We believe our results reveal the fundamental properties of the decision process made by the deep reinforcement learning policies, and can help in constructing safe, reliable and value-aligned deep neural policies.

## 1 Introduction

Reinforcement learning algorithms leveraging the power of deep neural networks have obtained state-of-the-art results initially in game-playing tasks Mnih et al. (2015) and subsequently in continuous control Lillicrap et al. (2015). Since this initial success, there has been a continuous stream of developments both of new algorithms Mnih et al. (2016), Hasselt et al. (2016), Wang et al. (2016), and striking new performance records in highly complex tasks Silver et al. (2017). While the field of deep reinforcement learning has developed rapidly, the understanding of the representations learned by deep neural network policies has lagged behind.

The lack of understanding of deep neural policies is of critical importance in the context of the sensitivities of policy decisions to imperceptible, non-robust features. Beginning with the work of Szegedy et al. (2014), Goodfellow et al. (2015), deep neural networks have been shown to be vulnerable to adversarial perturbations below the level of human perception. In response, a line of work has focused on proposing training techniques to increase robustness by applying these perturbations to the input of deep neural networks during training time (i.e. adversarial training) Goodfellow et al. (2015) Madry et al. (2017). However, there are also concerns about adversarial training including decreased accuracy on clean data Bhagoji et al. (2019), prohibiting generalization (e.g. a performance gap between adversarial and standard training that increases as the size of the dataset grows Chen et al. (2020)), and incorrect invariance to semantically meaningful changes Tramèr et al. (2020).

In our paper we focus on understanding the learned representations and policy vulnerabilities and ask the following questions: (i) *How do non-robust directions on the deep neural policy manifold interact with each other temporally and spatially?* (ii) *Do the non-robust features learnt by the deep reinforcement learning algorithms transform under adversarial attacks?* (iii) *How do these learnt correlated non-robust features change with distributional shift?* (iv) *Does adversarial training solve the problem of learning correlated non-robust features?* To be able to answer these questions in our paper we focus on understanding the representations learned by deep reinforcement learning policies and make the following contributions:

- We propose a novel tracing algorithm to analyze the spatially and temporally correlated vulnerabilities of deep reinforcement learning policies.

- We conduct several experiments in the Arcade Learning Environment with policies trained in high-dimensional state representations.

- We go over several benchmarked adversarial attack techniques and show how these attacks affect the vulnerable learned representations.

- We inspect the effects of distributional shift on the correlated non-robust feature patterns learned by deep reinforcement learning policies.

- Finally, we investigate the presence of non-robust features in adversarially trained deep neural policies, and show that the state-of-the-art adversarial training method leads to learning sparser and spikier vulnerable representations.

## 2 BACKGROUND AND PRELIMINARIES

### 2.1 PRELIMINARIES

A Markov Decision Process (MDP) is defined by a tuple $(\mathcal{S}, \mathcal{A}, \mathcal{P}, \mathcal{R})$ where $\mathcal{S}$ is a set of states, $\mathcal{A}$ is a set of actions, $\mathcal{P} : \mathcal{S} \times \mathcal{A} \times \mathcal{S} \to [0, 1]$ is a Markov transition kernel, $\mathcal{R} : \mathcal{S} \times \mathcal{A} \times \mathcal{S} \to \mathbb{R}$ is a reward function, and $\gamma$ is a discount factor. A reinforcement learning agent interacts with an MDP by observing the current state $s \in \mathcal{S}$ and taking an action $a \in \mathcal{A}$. The agent then transitions to state $s'$ with probability $\mathcal{P}(s, a, s')$ and receives reward $R(s, a, s')$. A policy $\pi : \mathcal{S} \times \mathcal{A} \to [0, 1]$ selects action $a$ in state $s$ with probability $\pi(s, a)$. The main objective in reinforcement learning is to learn a policy $\pi$ which maximizes the expected cumulative discounted rewards

$$R = \mathbb{E}_{a_t \sim \pi(s_t, \cdot)} \sum_t \gamma^t \mathcal{R}(s_t, a_t, s_{t+1}).$$

This maximization is achieved by iterative Bellman update

$$Q(s_t, a_t) = \mathcal{R}(s_t, a_t) + \gamma \sum_{s_t} \mathcal{P}(s_{t+1}|s_t, a_t) V(s_{t+1}) \tag{1}$$

Hence, the optimal policy $\pi^*(s, a)$ can be achieved by executing the action $a^*(s) = \arg\max_a Q(s, a)$ i.e. the action maximizing the state-action value function in state $s$.

### 2.2 ADVERSARIAL PERTURBATION TECHNIQUES AND FORMULATIONS

Following the initial study conducted by Szegedy et al. (2014), Goodfellow et al. (2015) proposed a fast and efficient way to produce $\epsilon$-bounded adversarial perturbations in image classification based on linearization of $J(x, y)$, the cost function used to train the network, at data point $x$ with label $y$. Consequently, Kurakin et al. (2016) proposed the iterative form of this algorithm named iterative fast gradient sign method (I-FGSM).

$$x_{\text{adv}}^{N+1} = \text{clip}_\epsilon(x_{\text{adv}}^N + \alpha \text{sign}(\nabla_x J(x_{\text{adv}}^N, y))). \tag{2}$$

This algorithm further has been improved by the proposal of the utilization of the momentum term by Dong et al. (2018). Following this Ezgi (2020) proposed a Nesterov momentum technique to compute $\epsilon$-bounded adversarial perturbations for deep reinforcement learning policies by computing

the gradient at the point $s_{\mathrm{adv}}^t + \mu \cdot v_t$,

$$v_{t+1} = \mu \cdot v_t + \frac{\nabla_{s_{\mathrm{adv}}} J(s_{\mathrm{adv}}^t + \mu \cdot v_t, a)}{\|\nabla_{s_{\mathrm{adv}}} J(s_{\mathrm{adv}}^t + \mu \cdot v_t, a)\|_1} \tag{3}$$

$$s_{\mathrm{adv}}^{t+1} = s_{\mathrm{adv}}^t + \alpha \cdot \frac{v_{t+1}}{\|v_{t+1}\|_2} \tag{4}$$

Another class of algorithms for computing adversarial perturbations focuses on different methods for computing the smallest possible perturbation which successfully changes the output of the target function. The DeepFool method of Moosavi-Dezfooli et al. (2016) works by repeatedly computing projections to the closest separating hyperplane of a linearization of the deep neural network at the current point. Carlini & Wagner (2017) proposed targeted adversarial formulations in image classification based on distance minimization between the original sample and the adversarial sample

$$\min_{x_{\mathrm{adv}} \in \mathcal{X}} c \cdot J(x_{\mathrm{adv}}) + \|x_{\mathrm{adv}} - x\|_2^2 \tag{5}$$

Another variant of this algorithm was proposed by Chen et al. (2018) based on $\ell_1$-regularization of the $\ell_2$-norm bounded Carlini & Wagner (2017) adversarial formulation.

$$\min_{x_{\mathrm{adv}} \in \mathcal{X}} c \cdot J(x_{\mathrm{adv}}) + \sigma_1 \|x_{\mathrm{adv}} - x\|_1 + \sigma_2 \|x_{\mathrm{adv}} - x\|_2^2 \tag{6}$$

## 2.3 Deep Reinforcement Learning Policies and Adversarial Effects

Beginning with the work of Huang et al. (2017) and Kos & Song (2017), which introduced adversarial examples based on FGSM to deep reinforcement learning, there has been a long line of research on both adversarial attacks and robustness for deep neural policies. On the attack side, the work of Lin et al. (2017) utilized the Carlini&Wagner method to compute perturbations in only a subset of states, strategically chosen to maximize impact. Ezgi (2020) showed Nesterov momentum produced adversarial perturbations that are faster to compute compared to Carlini & Wagner (2017) with similar or better impact on the policy performance. More intriguingly, the work of Korkmaz (2022) showed that deep reinforcement learning policies learn similar adversarial directions across MDPs intrinsic to the training environment. On the defense side Pinto et al. (2017) model the interaction between an adversary producing perturbations and the deep neural policy taking actions as a zero-sum game, and train the policy jointly with the adversary in order to improve robustness. More recently Huan et al. (2020) formalized the adversarial problem in deep reinforcement learning by introducing a modified MDP definition which they term State-Adversarial MDP (SA-MDP). Based on this model the authors proposed a theoretically principled *certified robust* adversarial training algorithm called SA-DQN. While recent studies raised some concerns on the drawbacks of adversarial training techniques Korkmaz (2021) in terms of limiting the generalization capabilities of the trained policy, these studies lack a method of explaining and understanding the main problems of the adversarial training techniques.

## 3 Principal High-Sensitivity Direction Analysis

In our paper our goal is to seek answers for the following questions:

- *How can we inspect the correlated non-robust features across time and space in deep reinforcement learning policies?*
- *How does distributional shift affect the vulnerabilities of learned representations?*
- *Do correlated non-robust features transform for policies under adversarial perturbations?*
- *Does adversarial training prohibit learning correlated non-robust features?*

To be able to answer these questions we propose a novel policy imaging technique that traces correlated high-sensitivity directions in the deep neural policy landscape across time. In the remainder of this section we explain in detail our proposed method.

**Definition 3.1.** Let $v \in \mathbb{R}^d$ be a vector with $\|v\|_2 = 1$. Let $Q$ be a state-action value function and for a state $s$ let $\hat{s} = s + \epsilon w$. For $\epsilon > 0$, the vector $v$ is an $\epsilon$-high-sensitivity direction for $Q$ in state $s$ if

$$v = \arg\max_{\|w\|_2=1} Q(\hat{s}, \arg\max_a Q(\hat{s}, a)) - Q(\hat{s}, \arg\max_a Q(s, a)). \tag{7}$$

In words, $v$ is a high-sensitivity direction if adding a perturbation of $\ell_2$-norm epsilon along $v$ maximizes the difference between the optimal state-action value in the new state and the value assigned in the new state to the previously optimal action. Eqn 7 can be approximated by using the softmax cross entropy loss. Hence, we define $\pi(s,a)$ as the softmax policy of the state-action value function.

$$\pi(s,a) = \frac{e^{(Q(s,a)/T)}}{\sum_{a' \in A} e^{(Q(s,a')/T)}}.$$

The cross entropy loss between the softmax policy in state $s_g$ and the argmax policy $\tau(s,a) = 1_{a = \arg\max_{a'} \pi(s,a')}(a)$ at state $s$ is

$$J(s,s_g) = -\sum_{a \in A} \tau(s,a) \log(\pi(s_g,a)) = -\log(\pi(s_g, a^*(s))).$$

Therefore by definition of the softmax policy we have

$$J(s,s_g) = \log \sum_{a' \in A} e^{Q(s_g,a')/T} - Q(s_g, a^*(s)) \approx Q(s_g, a^*(s_g)) - Q(s_g, a^*(s))$$

where the final approximate equality becomes an equality as $T \to 0$. Setting $v = s_g - s$, shows that maximizing the softmax cross entropy approximates the maximization in Definition 7. Hence, the gradient $\nabla_{s_g} J(s,s_g)|_{s_g=s}$ gives the direction of the largest increase in cross-entropy when moving from state $s$. Intuitively this is the direction along which the policy distribution $\pi(s,a)$ will most rapidly diverge from the argmax policy. Hence, $\nabla_{s_g} J(s,s_g)|_{s_g=s}$ is a high-sensitivity direction in the neural policy landscape in state $s$. Most importantly, the reason we trace high-sensitivity directions is to uncover the non-robust features learnt by the deep neural policy. To capture the correlated non-robust features we must aggregate the information on high-sensitivity directions from a collection of states visited while utilizing the policy $\pi$ in a given MDP. We thus define a single direction which captures the aggregate non-robust feature information from multiple states as follows:

**Definition 3.2.** Given a set of $n$ states $S = \{s_i\}_{i=1}^n$ the principal high-sensitivity direction is the vector $\mathcal{G}_S$ given by

$$\mathcal{G}_S = \arg\max_{\{v \in \mathbb{R}^d | \|v\|_2 = 1\}} \frac{1}{n} \sum_{i=1}^n \langle v, \nabla_{s_g} J(s_i, s_g)|_{s_g=s_i} \rangle^2.$$

**Proposition 3.1.** *Given a set of $n$ states $S = \{s_i\}_{i=1}^n$ define the matrix $\mathcal{L}(S)$ by*

$$\mathcal{L}(S) = \frac{1}{n} \sum_{i=1}^n \nabla_{s_g} J(s_i, s_g)|_{s_g=s_i} [\nabla_{s_g} J(s_i, s_g)|_{s_g=s_i}]^\top.$$

*Then $\mathcal{G}_S$ is the eigenvector corresponding to the largest eigenvalue of $\mathcal{L}(S)$.*

*Proof.* Observe that by linearity of the inner product

$$\frac{1}{n} \sum_{i=1}^n \langle v, \nabla_{s_g} J(s_i, s_g)|_{s_g=s_i} \rangle^2 = \frac{1}{n} \sum_{i=1}^n v^\top \nabla_{s_g} J(s_i, s_g)|_{s_g=s_i} [\nabla_{s_g} J(s_i, s_g)|_{s_g=s_i}]^\top v$$

$$= v^\top \left( \frac{1}{n} \sum_{i=1}^n \nabla_{s_g} J(s_i, s_g)|_{s_g=s_i} [\nabla_{s_g} J(s_i, s_g)|_{s_g=s_i}]^\top \right) v$$

$$= v^\top \mathcal{L}(S) v.$$

Thus $\mathcal{G}_S = \arg\max_{\{v \in \mathbb{R}^d | \|v\|_2 = 1\}} v^\top \mathcal{L}(S) v$. Therefore, by the variational characterization of eigenvalues, $\mathcal{G}_S$ is the eigenvector corresponding to the largest eigenvalue of $\mathcal{L}(S)$. $\square$

Intuitively, $\mathcal{G}_S$ is the direction that has the highest average correlation with the gradients of the states in $S$. Also note that $\mathcal{G}_S$ has the same dimensions as each state $s$, and so can easily be rendered in the same format as the states to visualize non-robust features. In Proposition 3.1 we show that $\mathcal{G}_S$ can be computed by solving an eigenvalue problem. Proposition 3.1 is the basis for Algorithm 1, which computes $\mathcal{G}_S$ by first calculating $\mathcal{L}(S)$ by summing over states, and then outputs the maximum eigenvector.

---

**Algorithm 1:** RADEN: Robustness Analysis via Dominant Eigendirections in Neural Manifold

---

**Input:** State-action value function $Q(s, a)$, actions $a \in \mathcal{A}$, states $s \in \mathcal{S}$, policy $\pi(s, a)$
**Output:** Principal high-sensitivity direction $\mathcal{G}(i, j)$
**for** $s = s_0$ **to** $s_T$ **do**
    $\tau(s, a) = 1_{a = \arg\max_{a'} Q(s, a')}(a)$
    $\pi(s_{\mathrm{g}}, a) = \mathrm{softmax}(Q(s_{\mathrm{g}}, a))$
    $J(s, s_g) = \sum_{a \in A} \tau(s, a) \log(\pi(s_g, a))$
    $\mathcal{L} \mathrel{+}= \nabla_{s_g} J(s, s_g)|_{s_g = s} [\nabla_{s_g} J(s, s_g)|_{s_g = s}]^\top$
**end for**
**Return:** Eigenvector $\mathcal{G}$ corresponding to largest eigenvalue of $\mathcal{L}$

---

Next we show how RADEN can be used to measure the effects of environment changes on the correlated non-robust features both visually and quantitatively. Let $S$ be the set of states encountered when utilizing policy $\pi$ in an MDP $\mathcal{M}$, and let $S'$ be the set of states encountered when utilizing $\pi$ where the environment in $\mathcal{M}$ is modified by applying some change to each state $s$. In this setting, comparing the visualization of $\mathcal{G}_S$ and $\mathcal{G}_{S'}$ can give a qualitative picture of how the environmental change affects the learned correlated non-robust features. In order to give a more quantitative metric for this change we define

**Definition 3.3.** For two sets of states $S$ and $S'$, the feature correlation quotient is given by

$$\Lambda(S', S) = \frac{\mathcal{G}_{S'}^\top \mathcal{L}(S) \mathcal{G}_{S'}}{\mathcal{G}_S^\top \mathcal{L}(S) \mathcal{G}_S}.$$

Next we show that the feature correlation quotient is bounded between zero and one.

**Proposition 3.2.** *For any two sets of states $S$ and $S'$*

$$0 \leq \Lambda(S', S) \leq 1.$$

*Proof.* By Proposition 3.1,

$$\mathcal{G}_{S'}^\top \mathcal{L}(S) \mathcal{G}_{S'} \leq \max_{\|v\|_2 = 1} v^\top \mathcal{L}(S) v = \mathcal{G}_S^\top \mathcal{L}(S) \mathcal{G}_S$$

Thus the numerator of $\Lambda(S', S)$ is always less than or equal to the denominator i.e. $\Lambda(S', S) \leq 1$. Furthermore, $\mathcal{L}(S)$ is positive semidefinite, as it is a sum of rank one projection matrices, and so $\Lambda(S', S) \geq 0$. □

Therefore, the feature correlation quotient $\Lambda(S', S)$ is a number between zero and one which intuitively measures how correlated the non-robust features from $S'$ are to those from $S$. When measuring how an environmental change affects non-robust features, it is also important to take the stochastic nature of the MDP into account. In particular, the non-robust features observed with two different executions of the same policy may differ slightly due to the inherent randomness of the MDP. To account for this, we first collect a baseline set of states with no modification $S_0$. We then collect a set of states $S_1$ with no modification, and $S_1'$ with modification. By comparing $\Lambda(S_1, S_0)$ to $\Lambda(S_1', S_0)$ we can see how much of the decrease in average correlation is caused by the stochastic nature of the MDP, and how much of the decrease is caused by the environmental change.

## 4 EXPERIMENTAL RESULTS

The deep reinforcement learning policies evaluated in our experiments are trained with the Double Deep Q-Network algorithm Hasselt et al. (2016) with the architecture proposed by Wang et al. (2016), and State-Adversarial Double Deep Q-Network (see Section 2.3) with experience replay Schaul et al. (2016). The set of states $S$ is collected over 10 episodes. The adversarial perturbation hyperparameters are: for the Carlini&Wagner formulation $\kappa$ is 10, learning rate is 0.01, initial constant is 10, for the elastic-net regularization formulation $\beta$ is 0.0001, learning rate is 0.1, maximum iteration is 300, for Nesterov Momentum $\epsilon$ is 0.001, and decay factor is 0.1. These hyperparameters



Figure 1: RADEN results of untransformed states and states under adversarial perturbations computed via Carlini&Wagner, Nesterov Momentum, and elastic-net regularization for Pong. Column1: Untransformed. Column2: C&W. Column3: Nesterov Momentum. Column4: Elastic-Net



Figure 2: RADEN results of untransformed states and states under adversarial perturbations computed via Carlini&Wagner, Nesterov Momentum, and elastic-net regularization for BankHeist. Column1: Untransformed. Column2: C&W. Column3: Nesterov Momentum. Column4: Elastic-Net

Table 1: The feature correlation quotient $\Lambda(S', S)$ in BankHeist, Freeway, RoadRunner, and Pong for the adversarial transformations: Carlini&Wagner, Nesterov Momentum, DeepFool, Elastic-Net.

| MDP | Untransformed | Carlini&Wagner | Nesterov | DeepFool | Elastic-Net |
|---|---|---|---|---|---|
| Freeway | 0.9917±0.0023 | 0.9499±0.02056 | 0.7868±0.02162 | 0.6869±0.02981 | 0.72590±0.0592 |
| BankHeist | 0.8360±0.0116 | 0.2837±0.02316 | 0.3407±0.02412 | 0.1748±0.04421 | 0.30917± 0.0521 |
| RoadRunner | 0.7652±0.0385 | 0.1621±0.02199 | 0.3826±0.03118 | 0.5353±0.03127 | 0.52506± 0.0782 |
| Pong | 0.4934±0.0391 | 0.0408±0.04056 | 0.3444±0.01981 | 0.3277±0.02871 | 0.10529± 0.0629 |

are consistent with prior work to achieve the most effective adversarial perturbations (i.e. perturbations causing the largest decrease on the discounted expected cumulative rewards obtained by the policy).

## 4.1 IMAGING NON-ROBUST FEATURE SHIFTS UNDER ADVERSARIAL PERTURBATIONS

In this section we investigate the effects of adversarial attacks on the learnt correlated non-robust features. Figure 2 shows the RADEN results for the untransformed states and the adversarially attacked states in the Pong and BankHeist environments. In particular, these perturbations are computed via the Nesterov momentum, Carlini&Wagner, and elastic-net regularization formulations (see Section 2.2). Figure 2 demonstrates that different adversarial formulations surface different sets of correlated non-robust features. Depending on the perturbation type, the correlated non-robust features can change quite noticeably. In fact, while the Carlini&Wagner formulation leaves a distinct signature on the vulnerable representation pattern, the non-robust features under Nesterov momentum appear most similar to those of the untransformed states. Our imaging technique can help to understand the vulnerabilities of deep reinforcement learning policies by allowing us to visualize precisely how non-robust features change with different perturbations.

Table 1 shows the feature correlation quotient $\Lambda(S', S)$ results where $S$ consists of untransformed states and $S'$ consists of states modified by the Nesterov Momentum, Carlini&Wagner, elastic-net regularization and DeepFool formulations respectively. Note that in all games the setting where $S'$ consists of a set of untransformed states from an independent execution has the highest feature correlation quotient $\Lambda(S', S)$. Therefore the additional decrease of $\Lambda(S', S)$ when $S'$ is modified by adversarial perturbations can be attributed to changes in non-robust features caused by the perturbations. Observe also that the qualitative similarity between the visualizations in Figure 2 of the different transformed states is matched by their ranking under $\Lambda(S', S)$ i.e. sorting from largest to smallest correlation quotient for BankHeist yields Nesterov momentum, elastic-net, and then Carlini&Wagner. The fact that the feature correlation quotient has distinct results for untransformed states and for states under all the types of adversarial formulations indicates that RADEN can facilitate detecting different types of adversarial perturbations.

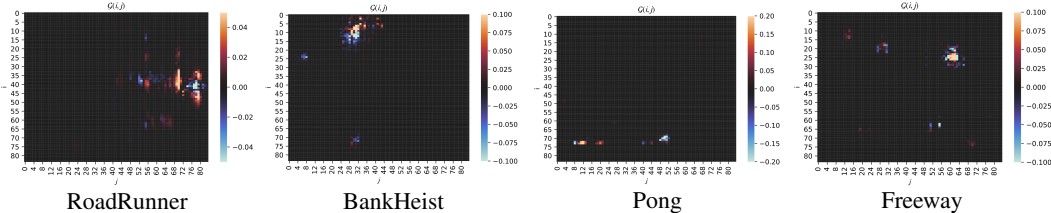

Figure 3: RADEN results of the state-of-the-art adversarially trained deep reinforcement learning policies for BankHeist, Pong, Freeway and RoadRunner.

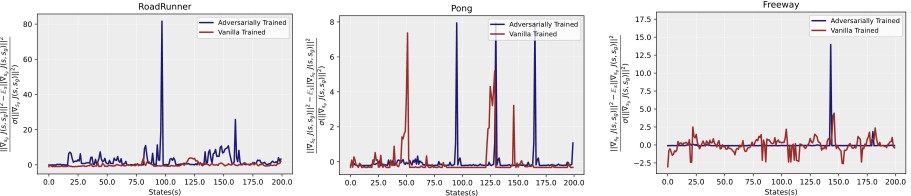

Figure 4: Standardized gradients $\|\nabla_{s_g} J(s_i, s_g)\|^2$ for vanilla trained and state-of-the-art adversarially trained deep reinforcement learning policies.

## 4.2 EXISTENCE OF CORRELATED NON-ROBUST FEATURES IN ADVERSARIAL TRAINING

In this section we investigate the effects of adversarial training on the correlated non-robust features. In particular, the SA-DDQN algorithm adds the regularizer $\mathcal{R}$,

$$\mathcal{R}(\theta) = \sum_s \left( \max_{\bar{s} \in D_\epsilon(s)} \max_{a \neq a^*(s)} Q_\theta(\bar{s}, a) - Q_\theta(\bar{s}, a^*(s)) \right).$$

during training in the temporal difference loss. Figure 3 shows the RADEN results for the state-of-the-art adversarially trained deep reinforcement learning policies. The non-robust features of the adversarially trained deep reinforcement learning policies are much more tightly concentrated on a small number of coordinates in the state observations, and these areas of concentration have moved significantly from where they were under vanilla training. Thus, the visualization allows us to see that correlated, non-robust features persist in adversarially trained policies, albeit in different locations with sparser patterns than vanilla trained deep reinforcement learning policies.

In Figure 6 we plot the Fourier transform of $\mathcal{G}_S$ where $S$ is collected from a vanilla trained and an adversarially trained policy in RoadRunner, BankHeist, Pong and Freeway. The Fourier transform reveals clear differences in the spatial frequencies occupied by $\mathcal{G}_S$ under vanilla and adversarial training. There is a consistent trend that the larger entries of the Fourier transform are more evenly and smoothly spread out for the adversarially trained policies. Thus, adversarial training leaves a consistent signature on the non-robust features detectable via the Fourier transform of $\mathcal{G}_S$. There is also a change in orientation: if the larger entries of the Fourier transform for the vanilla trained policy are more spread out along one axis, the adversarially trained Fourier transform is more spread along the other.

To complete our analysis of adversarial training we further include results on how non-robust features vary across time. To do so we record the $\ell_2$-norm of the gradient $\|\nabla_{s_g} J(s_i, s_g)\|^2$ in each state $s_i \in S$ for both adversarially trained and vanilla trained policies in RoadRunner, Pong, and Freeway. The results are plotted in Figure 4. In both RoadRunner and Freeway, the adversarially trained policy has relatively small gradients except for a few dramatic spikes corresponding to much higher sensitivity. This is in contrast to the vanilla trained policy in these games which tends to have a much smoother distribution which remains closer to the mean. These results indicate that adversarial training introduces higher jumps in sensitivity over states (i.e. extreme sensitivity for a new set of states) when compared to vanilla training.

## 4.3 THE EFFECTS OF DISTRIBUTIONAL SHIFT ON THE NON-ROBUST FEATURES LEARNT

To evaluate the effects of distributional shift on the learnt policy we go over several environment modifications with RADEN. These transformations include blurring, perspective transform, rotation,

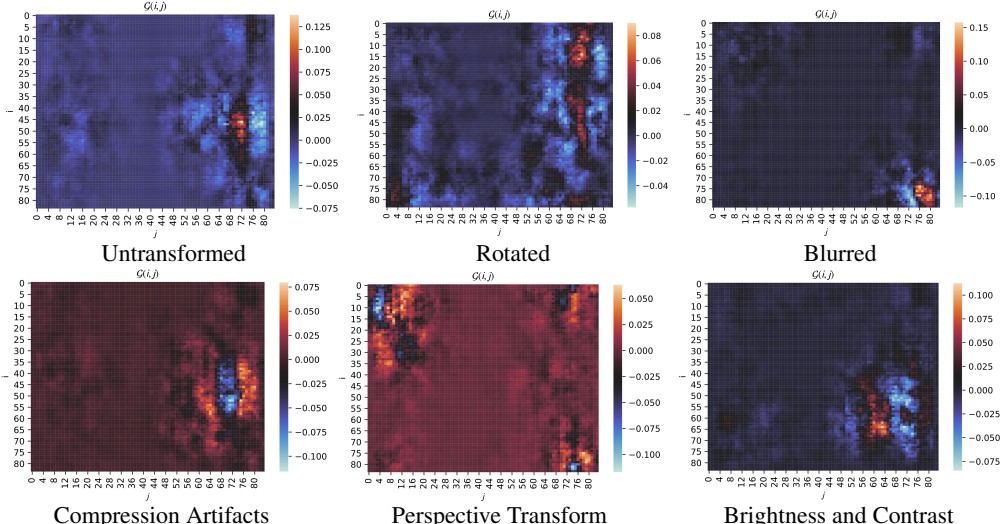

Figure 5: RADEN results of untransformed states and states under natural transformations with rotation, perspective transformation, blurring, compression artifacts, and brightness and contrast for Pong.

Table 2: The feature correlation quotient $\Lambda(S', S)$ in BankHeist, Freeway, RoadRunner, and Pong for the natural transformations: brightness and contrast, compression artifacts, rotation modification, perspective transform, blurred observations.

| Distributional Shift | Freeway | BankHeist | RoadRunner | Pong |
|---|---|---|---|---|
| Untransformed States | 0.9917±0.0023 | 0.8360±0.0116 | 0.7652±0.0385 | 0.4934±0.0391 |
| Brightness and Contrast | 0.86756±0.0271 | 0.3095±0.0429 | 0.4369±0.0334 | 0.1678±0.0427 |
| Compression Artifacts | 0.90564±0.237 | 0.38814±0.022 | 0.24358±0.0204 | 0.49341±0.0191 |
| Rotation Modification | 0.1381±0.0081 | 0.2951±0.0062 | 0.3350±0.0050 | 0.13648±0.0032 |
| Perspective Transform | 0.3010±0.0281 | 0.1723±0.0311 | 0.3308±0.0274 | 0.4278±0.0196 |
| Blurred Observations | 0.2657±0.0148 | 0.0954±0.0127 | 0.2496±0.0162 | 0.0847±0.0083 |

brightness, contrast, and compression artifacts.[1] To keep these transformations small we bound by the LPIPS metric proposed by Zhang et al. (2018). Figure 5 shows the visualizations of $\mathcal{G}_S$ for states $S$ collected under the six environment modifications mentioned above. For the untransformed setting the visualization of $\mathcal{G}_S$ clearly emphasizes the center of the region where the agent's paddle moves up and down to hit the ball. The components of $\mathcal{G}_S$ take larger positive values at the center of this region and transition to negative values along the boundary. A similar emphasis can be found for the case of compression artifacts, but with the signs reversed (i.e. the center of the region is negative and the boundary is positive). The other transformations exhibit larger changes in the regions emphasized in the visualization with perspective transform, blurring, rotation, and brightness and contrast causing the emphasized region to move to different locations.

Table 2 contains the values of $\Lambda(S', S)$ where $S$ is collected from an untransformed run and $S'$ is collected from each of the six different transformations. In every game the largest value of $\Lambda(S', S)$ occurs when $S'$ comes from an independent untransformed run, indicating that the additional decrease observed for $S'$ from transformed runs is caused by the respective environmental transformations. It is notable that in Pong the second highest value for $\Lambda(S', S)$ occurs for $S'$ collected with compression artifacts, as this corresponds precisely to the qualitative similarity between the regions emphasized in the visualization of $\mathcal{G}_S$ for untransformed and compression artifacts. Hence, the results for $\Lambda(S', S)$ help us to quantitatively understand the effects of the environmental transformations, while agreeing well with the qualitative results of the RADEN visualizations. While the RADEN visualizations give us semantically meaningful information on the non-robust features learnt by the deep neural policy, also provides a detailed understanding of how these non-robust features change under environmental modifications.

---

[1]In particular, blurring refers to median blur, brightness and contrast to linear transformation, and compression artifacts to diminution in high frequency components due to JPEG conversion.

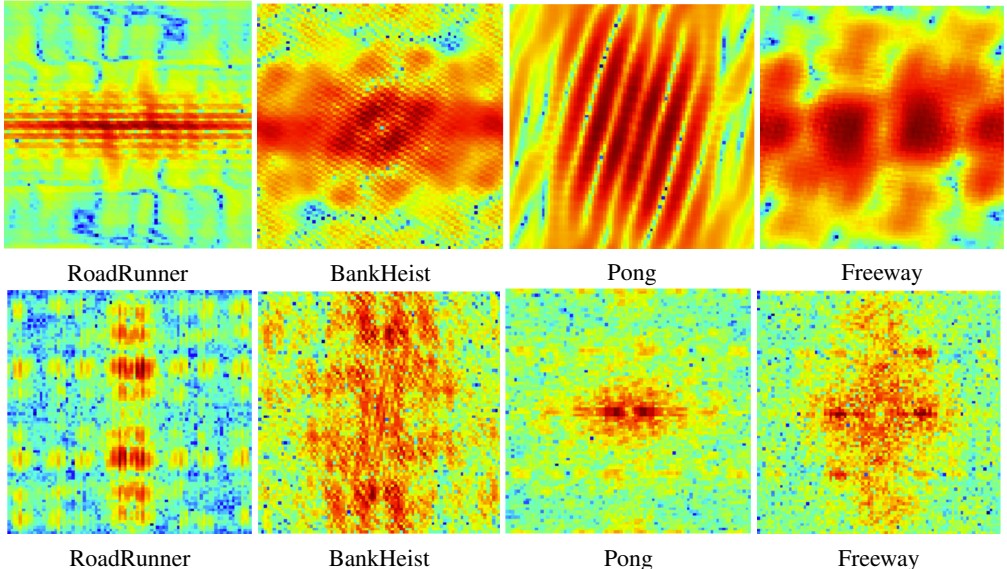

| RoadRunner | BankHeist | Pong | Freeway |

| RoadRunner | BankHeist | Pong | Freeway |

Figure 6: Fourier spectrum of the RADEN results of the state-of-the-art adversarially trained and vanilla trained deep neural policies. Row1: Adversarially trained. Row2: Vanilla trained.

The fact that RADEN can provide fine-grained vulnerability analysis of deep reinforcement learning policies under adversarial attacks, with distributional shift and with adversarial training can help with diagnosis of deep reinforcement learning policy vulnerabilities in the development phase. Conducting ablation studies with RADEN in reinforcement learning algorithm design can prevent building policies with embedded non-robust features, and our algorithm can be utilized to visualize and identify the effects of several design choices (e.g. algorithm, neural network architecture) on the non-robust features learnt by the policy from the MDP. In particular, given a visualization of the vulnerability pattern for a trained policy, one can try to modify the training environment in a way that will make the policy invariant to the non-robust features revealed by the visualization. Such modification could include changing the state representation in a way that does not change the semantics of the MDP, but does change the non-robust features in question. Furthermore, the effect of modifications to training algorithms can also be directly visualized, as exemplified by our results for adversarial training. Thus our method gives a straightforward way to diagnose or debug any proposed methods in terms of their effects on high-sensitivity directions.

## 5    CONCLUSION

In our paper we aim to seek answers for the following questions: (i) How do non-robust features learned by deep reinforcement learning policies correlate temporally and spatially? (ii) Do the correlated non-robust features remold under adversarial attacks? (iii) Does adversarial training result in policies that still end up learning non-robust features? (iv) How does distributional shift affect the learnt correlated non-robust features? To be able to answer these questions we analyze high-sensitivity directions in the deep neural policy landscape and we propose a novel tracing technique to detect correlated non-robust features learned by deep reinforcement learning policies. We show that deep reinforcement learning policies do end up learning correlated non-robust features, and that adversarial attacks lead to surfacing a new set of non-robust features or highlighting the existing ones. Most importantly, we show that the state-of-the-art adversarial training techniques also end up learning temporally and spatially correlated non-robust features. Finally, we show that distributional shifts introduce different sets of correlated non-robust features compared to adversarial attacks. Hence, our analysis not only allows us to effectively visualize correlated non-robust features, but also allows for precise understanding of changes in the learnt non-robust features caused by different training algorithms and different methods for altering states. Thus, we believe that our analysis can be useful both in diagnosing the vulnerabilities of deep neural policies, and in designing algorithms to improve robustness.

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
