# OpenReview forum: "Revealing Dominant Eigendirections via Spectral Non-Robustness Analysis in the Deep Reinforcement Learning Policy Manifold"
_ICLR.cc/2023/Conference — Submitted to ICLR 2023_

### Official Review · Reviewer_dkrX · 2022-10-23

**Confidence:** 4
**Clarity, Quality, Novelty And Reproducibility:** The presentation is poor.
**Correctness:** 2
**Technical Novelty And Significance:** 3
**Empirical Novelty And Significance:** 2
**Recommendation:** 3

**Strength And Weaknesses:**

Strengths And Weaknesses:
Pros: This paper studies a significant problem to understand the representations learned by deep neural network policies. The method proposed, as far as I know, is the first one to introduce a tracing technique to detect correlated non-robust features. The overall idea is interesting and innovative.
Cons:
	(1) I think this paper may need lots of improvement in its writing. It's informal to just use Eqn to refer to the equation, such as Eqn(5) rather than Equation (5) or Eq. (5). There are a lot of sentences in the text that confuse me.
	(2) The title of the paper mentions the use of spectral analysis, but the methods section does not clearly explain how to use spectral analysis.
	(3) The experimental section does not have an introduction to the experimental setup.
Questions:
	(4) The title of the paper mentions policy manifold, but there is no definition of policy manifold in the context. Please give a clear explanation of the policy manifold.
	(5) Please give a clear Explanation of the relationship between the learned representations of neural network policies and policy manifold.
	(6) What is the distributional shift in section 3 question 2?
	(7 )Give an explanation of ε and ω in Definition 3.1.
	(8) Give a clear explanation of the experimental graph, including the meaning of the coordinate axes, etc.


**Summary Of The Paper:**

This work analyzes high-sensitivity directions in the deep neural policy landscape and proposes a novel tracing technique to detect correlated non-robust features learned by deep reinforcement learning policies. It firstly defines $v$ as a high-sensitivity direction from state $s$ and $\hat s$ and then approximates $v$ by using the softmax cross entropy loss between the softmax policy and the argmax policy. It also proposes the RADEN algorithm to measure the effects of environmental changes on the correlated non-robust features visually and quantitatively by the Eigenvector G corresponding to the largest eigenvalue of L for the given state set S. Experiments on the Arcade Learning Environment are done to evaluate the idea.


**Summary Of The Review:**

The main body of the paper is far away from the title and claimed idea.

---

### Official Review · Reviewer_B6vV · 2022-11-01

**Confidence:** 2
**Correctness:** 2
**Technical Novelty And Significance:** 2
**Empirical Novelty And Significance:** 2
**Recommendation:** 3

**Clarity, Quality, Novelty And Reproducibility:**

The overall writing is good. But more explanations should be added to certain parts, or it could cause some difficulties in understanding the paper.

**Strength And Weaknesses:**

Some questions or suggestions about the paper:

1. The authors should address more why Definition 3.1 is an important notion that we should care about. And the formulation of $J(s, s_g)$ also seems to be arbitrary. It doesn’t make sense to let one policy to be the softmax policy and the other be the argmax policy. I guess the author is trying to show that in this setting, the gradient of $J$ is equivalent to Definition 3.1, but the equation (after the sentence “Therefore by definition of the softmax policy we have”) is clearly problematic, missing the factor $T$ somewhere. Even after correcting the equation, the formulation of $J$ still doesn’t make sense, one can just let both policies be softmax and still get a similar relation. Furthermore, wouldn’t it be much simpler to directly define $v$ to be the gradient of some metric function $J$?

2. The authors further define the notion $G_S$ given a set of states. However, the meaning of $G_S$ is also unclear. I would expect each state has its own sensitive direction, which may be of some interest for studying the instability of the policy under each state. But I don’t understand why the seemingly “averaged” vector $G_S$ still conveys the information that worth studying.

3. Propositions 3.1 and 3.2 are straightforward facts, and could be stated more concisely. And the boundedness condition of $\Lambda$ seems to be somehow unimportant.

4. The authors should give a more formal definition on what is a non-robust feature. According to my understanding, it refers to the high-sensitivity direction $G_S$, though I don’t know why it should be called a feature. $G_S$ is clearly a vector in the raw state space, whose dimension could be prohibitively large for the downstream computation.

5. The figures are hard to read and understand. While Figure 1 and Figure 2 are both RADEN results, in the text only Figure 2 is referred to. In Figure 1, the legend of the color is inconsistent across each subfigure. And the author should explain the meaning of these figures, does the highlighted areas imply the coordinates that are more vulnerable?

6. The authors claim the non-robust features still exist in adversarial training. They should given more explanations how they get this conclusion. The direction $G_S$ always exist, because any symmetric matrix always has the eigenvector with largest eigenvalue. Maybe it’s better to explain which shapes of $G_S$ indicate there exist non-robust features and which do not?

**Summary Of The Paper:**

The paper introduces a method to analyze the vulnerabilities of deep reinforcement learning policies. Experiments are conducted to illustrate the non-robust features, and show how adversarial attack techniques and robust training affect these features.

**Summary Of The Review:**

The authors propose a new method to study the non-robustness of the DRL policy. I suggest the authors add more explanations to their method and results. Instead of directly giving the definition of the notions $G_S$ and $\Lambda$, maybe it's better to address why these notions are important for understanding the DRL policy, and how they can be used to guide robust training.

---

### Official Review · Reviewer_iSgv · 2022-11-01

**Confidence:** 3
**Correctness:** 3
**Technical Novelty And Significance:** 2
**Empirical Novelty And Significance:** 2
**Recommendation:** 3

**Clarity, Quality, Novelty And Reproducibility:**

- I found the paper quite hard to read, which may have to do with the writing style of the authors. I also feel that even though most parts were clear, there were some parts that were not adequately justified (see the weakness section above).
- Given the paper is not very theoretical, I believe it would have benefitted a lot from a more extensive experimental evaluation, where the authors would show with specific examples how the improved vulnerability analysis stemming from their framework can lead to improved RL.
- Overall, I do not dispute the quality of the experiments and the authors have put effort in generating the various heatmaps. However, as mentioned previously, additional experiments could have been much more convincing about the utility of the proposed framework.
- I think the ideas in the paper are fairly novel (e.g., the high-sensitivity directions), but not significantly novel or disruptive. There are interesting ideas though.
- I believe the work can be reproduced - the authors provide details about their experiments and several detailed plots and tables. It would of course help if the authors release their code, after their work has been accepted.

**Strength And Weaknesses:**

Strengths
- The problem of determining the non-robust features in the neural policy manifold is very important with several potential applications.
- The definition of high-sensitivity directions is well-motivated and intuitive. The connection to the softmax cross entropy is also interesting.
- The authors complement the average high-sensitivity direction with the feature correlation quotient, and investigate how the two notions are related in their empirical analysis.
- The authors provide an empirical study of how correlated non-robust features also arise under adversarial attacks, adversarial training or distribution shift, and how they differ from those in vanilla training.

Weaknesses
- I am not clear how the paper shows that the direction with the highest average correlation can be so useful in practice. I am convinced about the usefulness of high-sensitivity directions at a given state. But I feel that the authors have not explained with convincing arguments the necessity and usefulness of the average high-sensitivity direction and the corresponding feature correlation quotient. For instance, assume we do RL. At a given time step, we would naturally be interested in the non-robust features and try to mitigate their impact; note that these may vary from one timestep to another. The average quantity certainly deserves merit, but I believe the authors should have motivated it further.
- I feel that a large part of the experimental analysis is about investigating how the proposed qualitative and quantitative metrics perform under vanilla training, as well as under adversarial attacks, adversarial training, or distributional shift. The findings are for sure interesting. But the question that was left unanswered: how can we leverage all these findings to improve RL? The authors devote one paragraph at the end of Section 4 to discuss how their framework can in principle provide fine-grained vulnerability analysis, but it would have made more sense in my opinion to show in practice rather than in theory how the introduced concepts can improve RL through improved vulnerability analysis (e.g., in a given MDP). I am concerned that much of this discussion has a very theoretical character.
- Some claims are not fully explained. As an example, the authors claim that there is a change in orientation in the Fourier transform: if the larger entries of the Fourier transform for the vanilla trained policy are more spread out along one axis, the adversarially trained Fourier transform is more spread along the other. This is true in the figure, but I am not sure that an adequate explanation was given.
- I did not find the contributions to be particularly novel - I liked the definition of high-sensitivity directions but I have some concerns with the treatment for average high-sensitivity directions.

**Summary Of The Paper:**

The paper introduces a method to determine the correlated high-sensitivity directions  in the deep neural policy manifold across space and time in the context of reinforcement learning. In this direction, they first define the concept of a high-sensitivity direction for the Q-function at a given state, and show that it is equal to the gradient of some cross-entropy loss at that state. The high-sensitivity direction will contain non-robust features, along which the Q-function changes the most. To capture the correlated non-robust features, the authors then propose to aggregate the information on high-sensitivity directions from a collection of states visited under some given policy. In particular, they suggest to use the direction with the highest average correlation with the gradients of all visited states, and show how this direction can be computed with a spectral algorithm. In addition, they introduce a feature correlation quotient, which captures how correlated the non-robust features from one set of states S' are to a different set of states S.

Based on these contributions, the authors then empirically investigate various questions. First, the effect of adversarial attacks on the learned correlated non-robust features, where they show that these can change quite significantly under adversarial perturbations. Second, the effects of adversarial training on the correlated non-robust features, where they show that they are more tightly concentrated in a small number of coordinates but in different locations that under vanilla training. Third, the effects of distributional shift, where they show that different transformations will change the location of the regions of correlated non-robust features.

**Summary Of The Review:**

Overall, I believe that this is a work with interesting ideas and experiments. However, I am not convinced that is ready for publication in its current form; I think that was is mainly missing is some empirical "proof" that the proposed framework can really improve RL via a better fine-grained vulnerability analysis, e.g., through concrete examples. The authors claim repeatedly that that this is the case, but, given the paper is not so theoretical, I would have expected a more through empirical evaluation where the authors clearly demonstrate the benefit of the proposed methodology.

---

### Official Review · Reviewer_eHD9 · 2022-11-01

**Confidence:** 4
**Correctness:** 2
**Technical Novelty And Significance:** 2
**Empirical Novelty And Significance:** 2
**Recommendation:** 3

**Clarity, Quality, Novelty And Reproducibility:**

There is a technical issue in the third display on page 4 in deriving $J(s, s_g)$. By sending $T \to 0$, $J(s, s_g)$ does not converge --- missing a pre-factor $T$.

Both Proposition 3.1 and Proposition 3.2 are elementary and direct consequences of their corresponding definitions. The proofs do not provide any insights either.

Figure 1 is not explained in the experiment section. It is also confusing because of inconsistent gauge of colors.

Figure 4 demonstrates that adversarial training and vanilla policy learning yield different Fourier spectrum in vulnerabilities. How do we understand the differences? Does it suggest that adversarial training is better, since it is relatively difficult to find attacks corresponding to the spikes in the Fourier spectrum?



**Strength And Weaknesses:**

================= Strength =================

Experimental study is comprehensive and provides many graphical illustrations.

The focus on the robustness of policies in reinforcement learning is of importance to many applications.

================= Weakness =================

Although RADEN helps to visualize high sensitivity directions, it is unclear how to incorporate such information for better design policy learning algorithms. In other word, practical implication of RADEN seems to be limited.

If I were not missing anything, spatial and temporal patterns of vulnerabilities in learned policies are very vaguely discussed, especially the temporal part. I don't think collecting states along a trajectory is an equivalence to temporal dependencies, as the vulnerabilities found by RADEN do not have temporal or even spatial information. The contributions are a bit overclaimed.

There lacks a coherent motivation of why studying the proposed four questions. The transition from the second paragraph to the third paragraph in Introduction is rather abrupt.

**Summary Of The Paper:**

The paper studies how to visualize high sensitivity directions in reinforcement learning. Given a set of states, authors proposed a RADEN algorithm to find the leading eigenvector of certain covariance matrix relevant to the perturbation analysis. The obtained leading eigenvector indicates the directions of vulnerability. Plenty of experimental results are presented to support the effectiveness of the proposed algorithm as well as extend to broader context of interest in reinforcement learning, e.g., distributional shift.

**Summary Of The Review:**

I think the manuscript in its current form needs substantial revision to meet the acceptance standard. My major concern is the unclear implications of the proposed method.

---

### Official Review · Reviewer_gVJV · 2022-11-01

**Confidence:** 4
**Correctness:** 3
**Technical Novelty And Significance:** 2
**Empirical Novelty And Significance:** Not applicable
**Recommendation:** 3

**Clarity, Quality, Novelty And Reproducibility:**

-The paper is written poorly and hard to follow.
- The novelty is not clear from the description in the paper.

**Strength And Weaknesses:**

*Strengths*
- The authors are considering an interesting problem of understanding of the representations learned by DNN policies in RL. This is of sufficient interest to the community.

*Weakness*

- What is non-robust direction? It is used on page 2 without clear definition/explanation.
- The motivation is not clear to me from the introduction. The authors started talking about the adversarial training issues in the standard supervised learning and then directly mentioned the challenges in deep reinforcement learning. A little more context and motivation is needed.
- There is no sufficient-related work to contrast the contributions with respect to the literature.
- Sec 2.2. and 2.3 seems redundant.
- In definition 3.1, is v and w are the same?
- The constraint in (7) is the equality constraint, can you comment if the contraint is convex, or if unique solution to problem in (7) exists?
- What is T in the definition of the policy after (7) ?
- "Eqn 7 can be approximated by using the softmax cross entropy loss."--why?
- How to calculate the  gradient of J with respect to state s? Would that require access to the true transition model P ?
- What is the meaning of non-robust and robust features with respec to the definition on 3.3?
- I am not sure what Figure 3 is describing?


**Summary Of The Paper:**

The authors have considered the problem of understanding the sensitivity of deep reinforcement learning representation landscape. The authors have proposed the tracing technique to detect correlated non-robust features. The authors have defined "high sensitive"  direction of Q function in Definition 3.1. This helps to define the high sensitivity direction in Definition 3.2, and eventually connected it to the eigenvector of the highest average correlation with the gradients of the states. The authors have provided experimental results as well in the paper.

**Summary Of The Review:**

The paper is difficult to follow. The ideas is interesting but the paper requires a lot of work before publication.

---

### Decision · Program_Chairs · 2023-01-20

**Decision:**

Reject

**Justification For Why Not Higher Score:**

The authors made no comment during the rebuttal.

**Justification For Why Not Lower Score:**

N/A

**Metareview: Summary, Strengths And Weaknesses:**

The authors aim to study the vulnerabilities of the representation learned in deep RL. In order to achieve this goal, the authors propose a tracing technique to detect correlated non-robust features.

According to the reviewers, the paper has a lot of issues in the clarity and writing. The paper also lacks reasonable justification of the proposed definitions of sensitivity. For example, one reviewer point out that "Spatial and temporal patterns of vulnerabilities in learned policies are very vaguely discussed, especially the temporal part. I don't think collecting states along a trajectory is an equivalence to temporal dependencies, as the vulnerabilities found by RADEN do not have temporal or even spatial information." None of the questions are answered by the authors.

The authors have not provided any justification of their work during the rebuttal. We suggest a clear reject.


**Summary Of Ac-Reviewer Meeting:**

N/A